# Defining Candidate Imprinted loci in *Bos taurus*

**DOI:** 10.3390/genes14051036

**Published:** 2023-05-02

**Authors:** Minou Bina

**Affiliations:** Department of Chemistry, Purdue University, West Lafayette, IN 47907, USA; bina@purdue.edu

**Keywords:** BCL6, bull spermatogenesis, cattle genomics, cattle muscle, cattle spermatogenesis, CNNM1, CNR1, farm animals, genome wide, genomic imprinting, SIX1, ZFP57, ZFBS-morph overlaps

## Abstract

Using a whole-genome assembly of *Bos taurus*, I applied my bioinformatics strategy to locate candidate imprinting control regions (ICRs) genome-wide. In mammals, genomic imprinting plays essential roles in embryogenesis. In my strategy, peaks in plots mark the locations of known, inferred, and candidate ICRs. Genes in the vicinity of candidate ICRs correspond to potential imprinted genes. By displaying my datasets on the UCSC genome browser, one could view peak positions with respect to genomic landmarks. I give two examples of candidate ICRs in loci that influence spermatogenesis in bulls: *CNNM1* and *CNR1*. I also give examples of candidate ICRs in loci that influence muscle development: *SIX1* and *BCL6*. By examining the ENCODE data reported for mice, I deduced regulatory clues about cattle. I focused on DNase I hypersensitive sites (DHSs). Such sites reveal accessibility of chromatin to regulators of gene expression. For inspection, I chose DHSs in chromatin from mouse embryonic stem cells (ESCs) ES-E14, mesoderm, brain, heart, and skeletal muscle. The ENCODE data revealed that the *SIX1* promoter was accessible to the transcription initiation apparatus in mouse ESCs, mesoderm, and skeletal muscles. The data also revealed accessibility of *BCL6* locus to regulatory proteins in mouse ESCs and examined tissues.

## 1. Introduction

For many centuries, *Bos taurus* populations were selected for economically important traits [1,2,3,4]. Examples include milk production [5,6,7,8] and beef quality: i.e., tenderness, juiciness, flavor, color, and fat content [9,10]. Furthermore, emerging novel approaches could expedite trait selection: examples include cloning, somatic cell nuclear transfer (SCNT), and assisted reproduction technologies (ART). However, such practices could produce developmental anomalies known as LOS: Large Offspring Syndrome [11,12,13,14,15,16,17,18]. Mechanistically, the abnormalities are often caused by defects in genomic imprinting—a process that impacts embryonic development in mammals [19,20,21,22,23]. Briefly, genomic imprinting directs repression of a subset of genes in either the maternal or the paternal autosomal chromosomes [20,24,25,26,27,28]. Insight into genomic imprinting emerged from studies on mice [20,26,29,30,31]. Subsequently, it was extended to studies of humans [32,33] and farm animals [3]. As in other mammals, the cattle genome includes imprinted domains, genes, and transcripts [3].

Genomic imprinting involves the orchestrated action of several enzymes and proteins [28]. The imprinting marks are epigenetic and consist of methylated CpGs in ICRs [34]. Also known as KAP1, TRIM28 is associated with several proteins required for imprinted gene repression [28,35,36]. The process is relatively complex. A subset of DNA methyltransferases (DNMTs) write the epigenetic marks in ICRs. ZFP57 binds clusters of methylated hexanucleotides (TGC^m^CGC) to protect the modified CpGs from demethylation [37,38,39,40]. SETDB1 trimethylates lysine 9 in histone H3 to produce repressive H3K9me3 marks in chromatin [29]. HP1 binds H3K9me3 marks to initiate chromatin condensation [28,41].

Since imprinted genes are important to developmental processes [19,20,21,22,23], it is crucial to innovate novel strategies for their identification. Efforts in that direction include trained statistical models [42]. Another approach applied algorithms to predict imprinting status of genes in the human genome [43]. Another method employed an array of SNPs (Single Nucleotide Polymorphisms) and usual molecular methods to identify epigenetic features that correlated with imprinting status in mice [44]. A different strategy applied weighting methods, machine learning, and data mining to characterize *Bos taurus* imprinted genes from genomic to amino acid attributes [45]. I have developed a genome-wide strategy that locates known, inferred, and candidate ICRs in mammals. My strategy assumes that genes in the vicinity of candidate ICRs are potential imprinted genes.

To date, I have applied my genome-wide studies to three different mammals: mouse [46], human [47], and *Bos taurus* [48]. In this report, I focus on demonstrating that my strategy could identify candidate ICRs for potential imprinted genes important to spermatogenesis in bulls, and myogenesis in *Bos taurus*. I also demonstrate that with Mouse ENCODE data [49], it is possible to obtain regulatory information about cattle.

## 2. Materials and Methods

A whole-genome assembly of *Bos taurus* [50] offered the opportunity to perform genome-wide analyses to discover ICRs in domestic cattle [48]. Briefly, from the UCSC genome browser, I downloaded the build bosTau8 assembly. For genome-wide studies, I created two text files. One file contained TGCCGC and its complement. The double-stranded DNA defines sequences that, after CpG methylation, bind the ICRs that depend on ZFP57 to regulate allele-specific gene expression [40]. The other file included the nucleotide sequences of unmethylated ZFBS-morph overlaps and their complementary sequences [51]. For details, see the results section. To produce one dataset, a Perl script determined the genomic positions of ZFBS-morph overlaps [52]. I used the same script to obtain the genomic positions of TGCCGC. To locate known and candidate ICRs, another Perl script scanned a specified chromosomal DNA to count the number of ZFBS-morph overlaps in a sliding window consisting of 850 bases. The script created a density plot by disregarding isolated overlap occurrences. This strategy eliminated background noise. After the steps above, I wrote UNIX subroutines to format the output files for display on the UCSC genome browser [53,54,55,56]. This resource is located at the University of California Santa Cruz (UCSC). It can be accessed at (https://genome.ucsc.edu/cgi-bin/hgGateway (accessed on 29 April 2023)).

## 3. Results

### 3.1. Defining the Positions of Known, Inferred, and Predicted ICRs in Bos taurus

To date, my strategy correctly pinpointed the genomic positions of a substantial fraction of fully characterized ICRs/gDMRs in three mammalian species [46,47,48]. The strategy is based on studies demonstrating that in emouse, nearly 90% of ICRs/gDMRs include composite DNA elements consisting of the ZFP57 binding site (ZFBS) overlapping a subset of the MLL1 morphemes [52]. I refer to these elements as ZFBS-morph overlaps. In these overlaps, the morphemes correspond to sites recognized by MLL1/KMT2A and MLL2/KMT2B [57,58]. The key feature of these sites is that they include two or more CpGs. MLL1 (Mixed-Lineage Leukemia 1) is the founding members of a protein family. Their structure contains a domain that contains methylates lysine 4 in histone H3 producing H3K4me3 marks in chromatin [59]. This H3 modification creates chromatin states that support gene expression [59,60]. Although it is not known whether the MLL1/MLL2 binding sites may contribute to allele-specific gene expression, their CpG-richness could be good targets for methylations of ICRs by DNA methyltransferases.

In ICRs, the hexameric ZFP57 binding sites are clustered [40]. It is well known that methylated CpGs tend to mutate to T as a consequence of deamination [61]. This disposition to mutations leads to CpG deficiency. However, although the nonmethylated hexamer (TGCCGC) includes a CpG, it is short and thus occurs frequently along chromosomal DNA (Figure 1). In contrast, ZFBS-morph overlaps include two or more CpGs. Therefore, their frequency is far less than that observed for TGCCGC along chromosomal DNA (Figure 1). Furthermore, in ICRs, the ZFBS-morph overlaps occur as clusters [52]. Therefore, by counting the number of these overlaps in a sliding window, it became possible to create density plots in which peaks identified cluster positions (Figure 1).

For ICR detection, I tailored my datasets for uploading on the UCSC genome browser to view peak positions in the context of genomic landmarks, including SNPs, CpG islands, genes, and transcripts. SNPs are important to genetic studies. Examples include accurate prediction of cattle breeds [62], association of body stature with frame size at puberty in cattle [63], and comprehensive characterization of loss of imprinting in LOS induced by assisted reproduction technologies [16]. On the browser, uploading the datasets reveals three custom tracks (Figure 1). With respect to selected genomic landmarks, a track displays the genomic positions of TGCCGC: the hexamer that after methylation binds ZFP57. A second track marks the positions of ZFBS-morph overlaps. A third track shows peak positions in the density plots (Figure 1). Peak positions are markers of known, inferred, and predicted ICRs along chromosomal DNA sequences [46,47,48].

The browser displays genomic data with respect to the underlying DNA sequence [64]. Therefore, the format of the datasets enables viewing results of genome wide analyses in the context of the sequences selected as reference. RefSeq Genes are known cattle genes; Non-Cow RefSeq Genes were deduced from comparative sequence analyses. Additionally, the browser offers built-in tools to display genomic information in dense, pack, or full formats [55,56]. For example, examine the peak positions in the plot obtained for the entire *Bos taurus* chromosome 9 (Figure 1). In this figure, peaks are shown in full format; dense format gives the location of RefSeq Genes, the CpG islands, ZFBS-morph overlaps, and the TGCCGC hexanucleotide. To identify genes in the vicinity of peaks, one could obtain closeup views. Subsequent sections give a few examples. Note that peak intensities vary and depend on the number of ZFBS-morph overlaps that they encompass. Evaluations have revealed that robust peaks encompass three or more ZFBS-morph overlaps. Peaks that cover two could be true or false positives. I annotated the genes manually. Along chromosome nine, manually annotated genes include *ZAC1*, *SMAP1*, *TENT5A*, *HACE1, CNR1, CITED2*, and *AFDN* (Figure 1). *ZAC1* is a known imprinted transcript in the *PLAGL1* locus in mice [65]. *ZAC1* is also an imprinted transcript in cattle [27]. The expression and the methylation pattern of *PLAGL1* are conserved in human and cattle [66]. In LOS induced by ART, *ZAC1* was biallelically expressed in fetuses [16]. In the *PLAGL1* locus in *Bos taurus*, a peak in plots is within the experimentally defined imprinted DMR [48]. Thus, my strategy correctly located an imprinted DMR in nearly 106 Mb long cattle chromosomal DNA (Figure 1).

Along *Bos taurus* chromosome 9, additional robust peaks correspond to candidate ICRs for potential imprinted genes (Figure 1). Among the annotated genes, *SMAP1* encodes a membrane-associated protein implicated in exerting a stimulatory effect on stroma-supported erythropoiesis [67]. Steady state erythropoiesis produces new erythrocytes [68]. TENT5A is a cytoplasmic poly(A) polymerase that regulates bone mineralization [69]. *TENT5A* knockout mouse displayed bone fragility and defects in skeletal mineralization [69]. TENT5A also impacts the formation of muscle fibers in adolescent idiopathic scoliosis by preserving production of myogenin [70]. HACE1 is an ankyrin containing E3 ubiquitin ligase [71]. Its deficiency decreased the number of synapses and caused both structural and behavioral neuropathologic features [72]. CITED2 is required for heart morphogenesis and the establishment of left-right axis during mouse development [73]. *CITED2*-null mice die during gestation with fully penetrant heart defects and partially penetrant laterality defects [73]. CITED2 is a transcription co-factor. It also contributes to sex determination and early gonad development [74]. The adherens junction formation factor (AFDN) is an F-actin scaffold protein with essential roles in the organization of cell-cell junctions of polarized epithelia [75]. In mouse, transcription of *AFDN* produces two major isoforms [75,76]. Their products were detected in distinguishable regions in the mouse central nervous system [75].

### 3.2. Examples of Candidate ICRs for Potential Imprinted Genes Important to Bull Fertility

In cattle, fertility is influenced by semen quality, testis size, and efficiency of sperm production [77]. As in other mammals, spermatogenesis in bulls is a highly regulated process that drives multiplication and differentiation of germ cells [78]. Spermatogenesis involves the differentiation of round spermatids into fully mature spermatozoa, in which the histones are replaced with cysteine-rich protamines to fully condense the DNA [79]. With the exception of humans, bulls have a lower efficiency of spermatogenesis than most other species [78]. In *Bos taurus* genomic DNA, my approach discovered a candidate ICR for the imprinted repression of *CNNM1* (Figure 2). In mouse testes, CNNM1 was produced from neonatal to adult stages, and was associated with cell cycle and differentiation of spermatogenic cells [80].

My approach also discovered a candidate ICR for the imprinted repression of *CNR1*: tetrahydrocannabinol receptor 1 (Figure 3). Endocannabinoids are naturally occurring lipids, and the principal component of marijuana [81]. They regulate a large array of physiological functions and behaviors. Since *CNR1* is expressed in elongating spermatids and spermatozoa, it affects male germ cell progression and sperm maturation mediated by endocannabinoids [82]. Although prevailing studies have emphasized the impact of endocannabinoids on the brain, this group of compounds also control reproductive events in males [81,82,83]. In *CNR1* knock-out mice, low levels of 17beta-estradiol affected chromatin remodeling in spermatid by interfering with high levels of DNA compaction by protamines [82].

### 3.3. Examples of Candidates ICRs for Genes That Impact Muscle Formation in Bos taurus

Skeletal muscle mass is an important economic trait [4]. In fact, several groups of farm animals are bred and selected to improve meat production [84]. The genesis of skeletal muscle involves complex mechanisms that control myogenesis at all stages of development [85]. Key regulators of myogenesis include SIX1, MYOD, and myogenin [4,86]. My predictive strategy discovered candidate ICRs for potential imprinted genes that impacted formation of muscles in *Bos taurus*. Examples include *SIX1* and *BCL6* (Figure 4 and Figure 5). SIX1 is the founding member of a transcription factor family whose structure includes a homeodomain for binding DNA [87]. SIX homeoproteins are produced in several tissues. Furthermore, this group of regulatory proteins impacts diverse differentiation processes [88]. Mice lacking *SIX1* died at birth due to malformations of several ribs. The mice also showed extensive muscle hypoplasia affecting most of the body muscles and certain hypaxial muscles [88]. Furthermore, SIX1 controls the expression of *MYOD* in adult muscle progenitor cells [86]. In cattle, *SIX1* was highly expressed in the longissimus thoracis [87]. Biochemical studies revealed that in bovine DNA, the *SIX1* promotor interacted with MYOD, PAX7, and CREB [87]. The other potential imprinted gene in cattle (*BCL6*) encodes another DNA-binding protein that controls gene expression [89,90]. In mice, at gestational day 17, *BCL6* was primarily expressed in skeletal muscle, the olfactory epithelium, and the epithelium lining the upper airways and esophagus [90].

### 3.4. Cell-Type and Tissue-Specific Expression Patterns of SIX1 and BCL6 in Mouse

To further demonstrate the power of my strategy, I also examined my datasets reported for mouse [51]. Two density plots show that as in *Bos taurus*, the mouse genome includes candidate ICRs for parent-of-origin-specific expressions of *SIX1* and *BCL6* (Figure 6 and Figure 7). As observed for *Bos taurus* (Figure 4), the candidate ICR for allele-specific S*IX1* repression is in a CpG island in mouse (Figure 6). In plots, a candidate ICR encompasses the longest *BCL6* transcriptional isoform in mouse (Figure 7). Next, I explored the effectiveness of ENCODE data for studies of cattle. For functional analyses of human and mouse genomes, the genome browser gives the option of viewing tracks, displaying large-scale experimental data produced by ENCODE: the Encyclopedia of DNA Elements [49,91,92]. The goal of the ENCODE Consortium was to discover and define the functional elements in mouse and human genomes [49,91]. Although the consortium has not applied their technologies to studies of cattle, one could assume that data reported for mice [49] could serve as a model for studies of other mammals. Therefore, I thought it could be informative to examine *SIX1* and *BCL6* loci in the context of ENCODE data obtained for mouse [49].

Among the extensive ENCODE data, I chose to display the DHSs in chromatin. Locating DHSs is a powerful tool to determine the accessibility of chromatin to regulators of gene expression [93,94,95]. DHSs mark the positions of regulatory DNA. They facilitate the discovery of all classes of *cis*-regulatory elements: i.e., promoters, enhancers, insulators, silencers, and locus control regions [96]. On the genome browser, I selected tracks displaying DHSs in chromatin prepared from mouse ESCs ES-E14, mesoderm, brain, heart, and skeletal muscle. The data revealed extensive DHSs across both the *Six1* and *Bcl6* loci (Figure 6 and Figure 7). In mouse ES-E14, one block of DHSs maps to the 5′ end of *SIX1* and extends to *Six1* intronic sequences (Figure 6). Thus, in ESCs, the *Six1* promoter is accessible to the transcription initiation apparatus. DHSs in intronic sequences and downstream of the gene correspond to DNA segments that upregulate or downregulate *Six1* expression in mouse ES-E14 (Figure 6). In the mesoderm, DHSs occur in several regions, indicating that the expression of *Six1* is regulated during mouse development (Figure 6). Lack of DHSs in the brain and the heart suggests that the *Six1* locus is not expressed in these tissues. Notably, DHSs punctate several regions in chromatin from mouse skeletal muscle (Figure 6). Thus, *Six1* is robustly expressed in mouse muscle tissues. Furthermore, the positions of DHSs are cell-type and tissue specific (Figure 6). Next, I examined the positions of DHSs in the *Bcl6* locus (Figure 7). These hallmarks of open chromatin structure are spread across the entire *Bcl6* locus in mouse ES-E14, mesoderm, brain, heart, and skeletal muscle. Notably, expression of *Bcl6* is very robust in skeletal muscle and controlled by numerous regulatory DNA segments (Figure 7).

## 4. Discussion

Genomic imprinting is an evolutionary novelty restricted to mammals [19,20,21,22,23,97,98]. In domesticated animals, genomic imprinting affects complex traits [99]. Since imprinted genes are expressed from either the maternal or the paternal allele [100,101], genomic imprinting reduces gene dosage from two to one. Since genomic imprinting impacts developmental processes, it is essential to identify the genes expressed in a parent-of-origin-specific manner [97]. Most imprinted genes were discovered based on various experimental criteria [22]. My strategy involved creating plots displaying the density of ZFBS-morph overlaps along chromosomal DNA [46,47,48]. Peaks in plots locate the genomic positions of ICRs that are ZFP57 dependent [46,47,48]. Across chromosome 9, peaks are almost fully resolved (Figure 1). This finding demonstrates that peak occurrences are rare events in genomic DNA. Furthermore, one could expect that peaks covering three or more ZFBS-morph overlaps occur less frequently than those that cover two (Figure 1). Overall, peaks in the density plots correctly located the ICRs in cattle imprinting domains, including *H19—IGF2*, *KCNQ1*, *IGF2R*, and *PEG3*. Peaks also located include: the essential ICR in the *GNAS* complex locus; and the ICRs in *PLAGL1*, *MEST*, *NNAT*, *MEG8*, *SNRPN*, *HERC3-NAP1L5*, and *INPP5F* loci [48]. Thus, it became increasingly plausible that in plots, additional peaks corresponded to candidate ICRs [48].

Previously, my strategy discovered candidate ICRs for several potential imprinted genes in *Bos taurus* [48]. Examples include *HMGA2* and *LCORL*. These genes affect body size and stature. A non-synonymous mutation in *HMGA2* decreased height in Shetland ponies and other small horses [102]. In rabbits, a deletion at the *HMGA2* locus caused dwarfism and altered craniofacial development [103]. In wild canids, SNPs in *LCORL* were associated with stature [104]. *Hmga2* knockout mice were impaired in muscle development and had diminished proliferation of myoblasts [105]. In contrast, overexpression of *Hmga2* boosted myoblast growth [105]. In fact, extreme size diversification is a hallmark reflecting domestication. For example: the *LCORL* locus was among the genes displaying strong signatures of selection characteristics of morphological changes in domesticated pigs [106]. Additionally, *HMGA2* and *LCORL* are among a few genes that have played major roles in the rapid evolution of animal size as a consequence of trait selection practiced by farmers and breeders [107].

For this report, I chose examples of candidate ICRs for potential imprinted genes important to farming industry: spermatogenesis in bulls and muscle development in cattle. *CNNM1* and *CNR1* impact spermatogenesis. A candidate ICR maps to the 5′ end of *CNNM1* (Figure 2). In mice, CNNM1 was associated with cell cycles and differentiation of spermatogenic cells [80]. The candidate ICR for *CNR1* is in a CpG island (CpG162) that encompasses the gene promotor (Figure 3). In *Cnr1* knock-out mice, low levels of 17beta-estradiol interfered with chromatin remodeling in spermatids [82]. CNR1 binds endocannabinoids [81]. They regulate numerous physiological processes including male reproductive events [81,82,83]. CNR1 functions during germ cell progression and sperm maturation mediated by the endocannabinoids [82]. In *Bos taurus*, two candidate ICRs map to genes that affect the development of muscles, and could therefore impact beef quality: *SIX1* and *BCL6* (Figure 4 and Figure 5). SIX1 is a transcription factor essential for embryonic myogenesis [86]. In muscle progenitor cells, SIX1 regulates the expression of *MyoD* [86]. In *Six1*-deficient mice, myogenesis was altered [88]. In cattle, the *SIX1* promotor interacted with several muscle-specific transcription factors [87]. These interactions could play key roles in SIX1-mediated skeletal muscle growth in cattle [87].

To further demonstrate the power of my approach, I showed that I could examine the *Six1* and *Bcl6* loci in the context of the ENCODE data and their imprinting status in mice (Figure 6 and Figure 7). As in *Bos taurus* (Figure 4), the candidate ICR for *Six1* maps to a CpG island in mouse genomic DNA (Figure 6). As in *Bos taurus* (Figure 5), the candidate ICR for *Bcl6* is at 5′ end of the longest *Bcl6* transcript (Figure 7). Furthermore, ENCODE provides experimental data that could be relevant to biochemical and developmental studies of cattle. Specifically, the ENCODE Consortium has mapped DHSs, transcription factor binding regions, chromatin modification patterns, and replication domains throughout the mouse genome in diverse cell and tissue types [49,108]. DHSs are signatures of chromatin accessibility to regulators of gene expression [109,110]. For example, bovine estrogen receptor bind to pre-existing nuclease hypersensitive sites in chromatin [111]. Furthermore, distal DHSs correspond to regulatory elements that upregulate or repress gene expression [109]. To conclude: my strategy locates known and candidate ICRs dispersed along *Bos taurus* chromosomal DNA sequences. The ENCODE data could reveal regulatory information about cattle.

## Figures and Tables

**Figure 1 genes-14-01036-f001:**
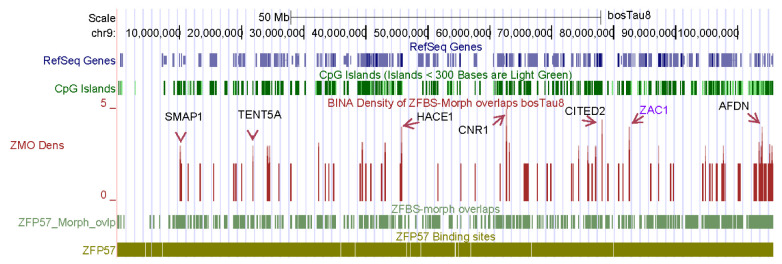
A snapshot of the density plot created for studies of chromosome 9. In this plot, a robust peak located the known ICR of *ZAC1* in nearly 106 Mb long chromosomal DNA. Additional robust peaks point to candidate ICRs near potential imprinted genes including *SMAP1*, *TENT5A*, *HACE1, CNR1, CITED2*, and *AFDN*.

**Figure 2 genes-14-01036-f002:**
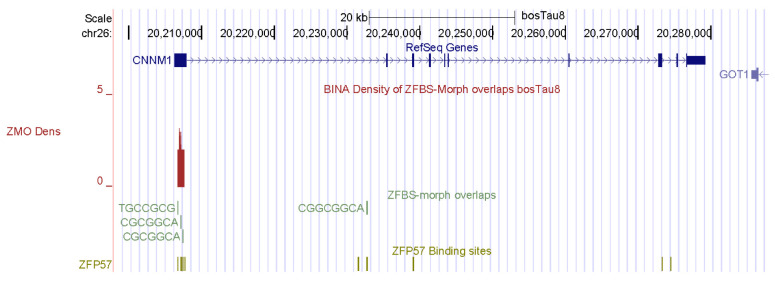
A candidate ICR at the 5′ end of *CNNM1*. In mice, *Cnnm1* triggered differentiation of spermatogenic cells.

**Figure 3 genes-14-01036-f003:**
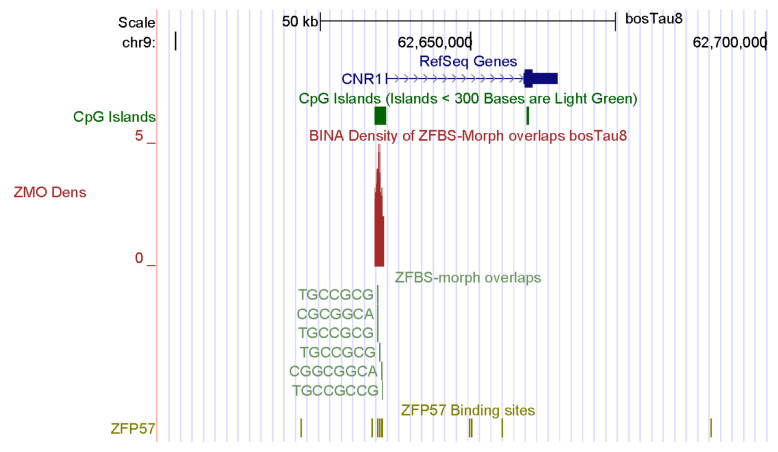
A candidate ICR at the 5′ end of *CNR1*. In mice, *CNR1* was expressed in elongating spermatids and spermatozoa, impacting male germ cell progression and sperm maturation mediated by the endocannabinoids.

**Figure 4 genes-14-01036-f004:**
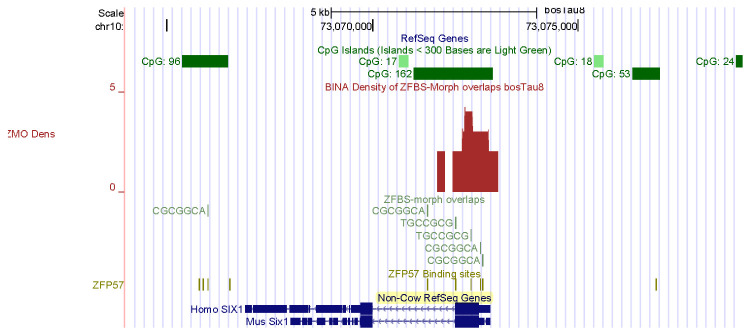
A candidate ICR at the 5′ end of *SIX1*. In adult muscle progenitor cells, SIX1 controls the expression of *MYOD*.

**Figure 5 genes-14-01036-f005:**
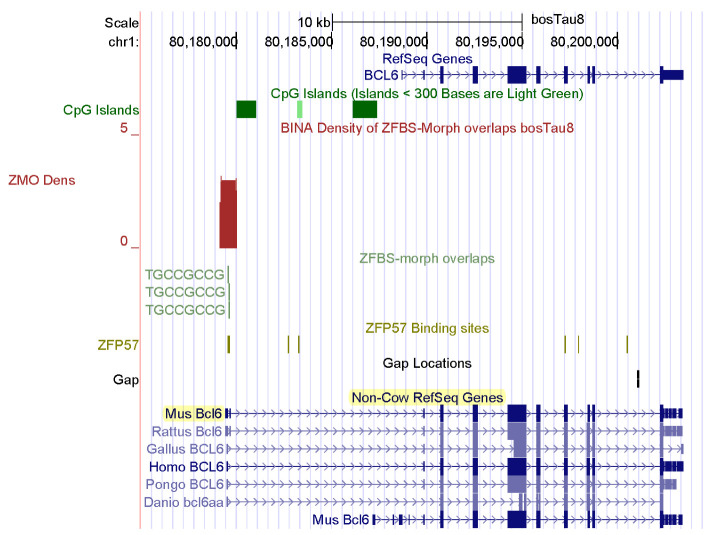
A candidate ICR at the 5′ end of *BCL6* with respect to Non-Cow RefSeq Genes. In mice, at gestational day 17, *BCL6* was primarily expressed in skeletal muscle.

**Figure 6 genes-14-01036-f006:**
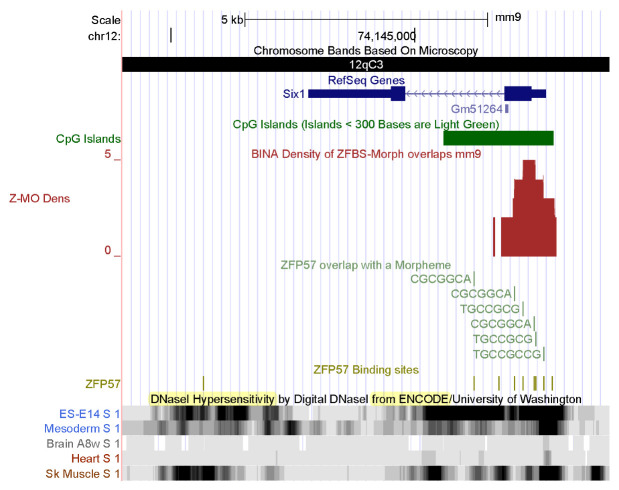
A candidate ICR at the 5′ end of *Six1* in mouse. Vertical bars reveal the positions of DHSs in chromatin prepared from mouse ESCs ES-E14, mesoderm, brain, heart, and skeletal muscle. Note cell-type and tissue-specific patterns of chromatin accessibility to regulatory proteins.

**Figure 7 genes-14-01036-f007:**
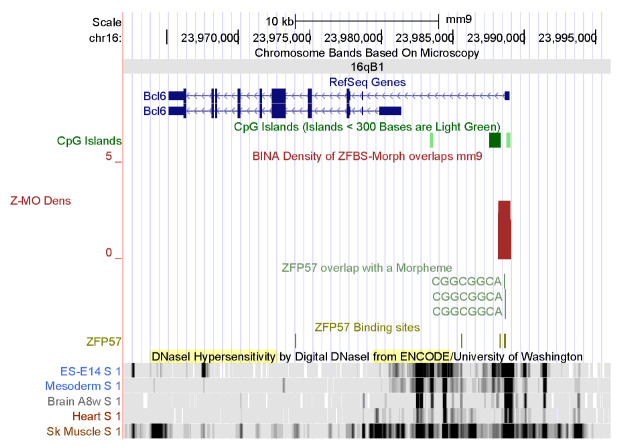
A candidate at the 5′ end of *Bcl6*in mouse. Several tracks display the positions of DHSs in chromatin prepared from mouse ESCs ES-E14, mesoderm, and skeletal muscle. Vertical bars mark the positions of DHSs in chromatin prepared from mouse ESCs ES-E14, mesoderm, brain, heart, and skeletal muscle. Note cell-type and tissue-specific patterns of chromatin accessibility to regulatory proteins.

## Data Availability

The data can be downloaded from the Purdue University Research Repository [112,113].

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
