# Peer review of "Defining Candidate Imprinted loci in Bos taurus"

_genes, 2023, doi:10.3390/genes14051036_

Reviewer 1 Report
This study entitled “Defining candidate imprinted loci in Bos taurus” show the genomic positions of candidate imprinting control regions of Bos taurus. This paper needs to be revised before accepting. Some sentences in the Materials and Methods and Results sections start with 'I' but should be written objectively. Also, in the Results, there is a mixture of what should be in the Materials and Methods and what should be in the Background and Discussion, which makes the paper difficult to read. The reviewer strongly recommends that the Results and the Discussion be presented in the same paragraph rather than separately, and write the Materials and Methods correctly (For example, “I refer to these elements as ZFBS-morph overlaps. (Line 88)” should be written in the Materials and Methods).
Major revision:
Line 29-30: Please provide the details of the development anomalies in farm cattle and how these impact on the producers' economy.
Line 62 – 66: The authors use “demonstrate” to summarize this study. However, this study used a database and did not prove it by actually knocking out the gene. The term is therefore considered inappropriate.
Minor revision
Abstract: What do UCSC, CNNM1, CNR1, SIX1, BCL6, and ENCODE stand for?
Line 25: Bos taurus should be italic.
Line 39-48: What do KAP1, TRIM28, HP1, DNMTs, ZFP57, SETDB1, and KRAB stand for?
Line 57-58: Is there any references to this sentence?
Line 73: What does ZFBS stand for?
Line 74: What is the version of Perl?
Line 84: What does DMR stand for?
Line 89: What do MLL and KMT stand for?
Line 140, 154, 159:, 213 What these genes do stand for?
Author Response
Thank you for your time spent on reviewing my manuscript. Below, with -* I have marked your
questions.
-* Some sentences in the Materials and Methods and Results sections start with 'I' but should be written
objectively……
Response: American style allows using I.
Also, in the Results, there is a mixture of what should be in the Materials and Methods.
Response: the issues you raise are stylistic. I have followed the style described in the guidelines.
-* Line 29-30: Please provide the details of the development anomalies in farm cattle and how these
impact on the producers' economy.
Response: Phenotypic manifestations include muscle and skeleton malformations, unusually bulky
bodies, and odd organ growth [18]. As detailed in [11-14], many calves die after birth. Loss of newly
born calves would reduce income that thus impact economy.
-* Line 62 – 66: The authors use “demonstrate” to summarize this study. However, this study used a
database and did not prove it by knocking out the gene. The term is therefore considered inappropriate.
Response: I did not use a database. I used my datasets computed for locating ICRs. Please see Data
Availability Statement: The data can be downloaded from the Purdue University Research Repository
[112, 113].
The goal of the research did not include knocking out genes. The goal was to predict the positions of
candidate ICRs genome wide. The results could help lab researchers to design experiments to discover
novel imprinted genes. Furthermore, the datasets could facilitate determining whether an
experimentally defined Differentially Methylated Region (DMR) corresponds to an ICR.
-* Abstract: What do UCSC, CNNM1, CNR1, SIX1, and BCL6, and ENCODE stand for?
Response: Located at the University of California Santa Cruz (UCSC), the browser provides genomic DNA
sequences of several species. It can be accessed at (h?ps://genome.ucsc.edu/cgi-bin/hgGateway).
The site also offers on-line resources for analyzing genomic DNA [53-56].
CNNM1, CNR1, SIX1, and BCL6 are approved gene names for including in publications. Below are
published definions:
CNNM1 (Cyclin and CBS Domain Divalent Metal Cation Transport Mediator 1),
CNR1 (Cannabinoid Receptor 1),
SIX1 (SIX homeobox 1),
BCL6 (Diffuse Cell Lymphoma 6),
SMAP1 (Small ArgGap1),
TENT5A (Terminal Nucleotidyltransferase 5A),
HACE1 (HECT domain and Ankyrin repeat Containing E3 ubiquitin-protein ligase 1),
CITED2 (CBP/p300 interacting transactivator with Glu/Asp rich carboxy-terminal domain 2),
AFDN (Afadin, Adherens Junction Formation Factor).
-* and what does ENCODE stand for?
Response: Encyclopedia of DNA Elements (ENCODE) [49, 88, 89]. The ENCODE Consortium has
integrated multiple technologies and approaches to discover and define the functional elements in
human and mouse genomic DNA sequences [49, 88].
You can access data produced by ENCODE at the UCSC genome browser. It might take some time to
learn how to use and view the data. It is worth the efforts.
Line 25: Bos taurus should be italic.
Response: OK. Thank you.
-* Line 39-48: What do KAP1, TRIM28, HP1, DNMTs, ZFP57, SETDB1, and KRAB stand for?
Response: KAP1 is an old name replaced by TRIM28 (Tripartite Motif Containing 28),
HP1 (HP1 Heterochromatin Protein 1).
DNMTs (DNA methyltransferases TRIM28),
ZFP57 (Zinc Finger Protein 57),
SETDB1 (SET domain bifurcated histone lysine methyltransferase 1),
KRAB (Kruppel-associated box).
For protein functions, please check reference [28].
Strogantsev R, Ferguson-Smith AC: Proteins involved in establishment and maintenance of imprinted
methylation marks. Brief Funct Genomics 2012, 11(3):227-239.
-* Line 57-58: Is there any references to this sentence?
Response: I have applied my genome wide studies to three different mammals: mouse [46], human [47],
and Bos taurus [48].
Line 73: What does ZFBS stand for?
Response: ZFP57 binding site [52].
It consists of a methylated double stranded hexamer ((TGCmCGC) recognized by ZFP57 in all ICRs and in
loci that could be ZFP57 dependent [40].
-* Line 74: What is the version of Perl?
Response: The latest version provided by the UNIX system at Purdue University.
-* Line 84: What does DMR stand for?
Response: Differentially Methylated Region.
Please note that all DMRs are not ICRs. My datasets could facilitate determining whether a DMR could
be an ICR.
Line 89: What do MLL and KMT stand for?
Response:
MLL1 (Mixed-Lineage Leukemia 1),
KMT (K lysine, MT methyltransferase).
The structure of MLL1 family contains a domain that methylates lysine 4 in histone H3 producing
H3K4me3 marks in chromatin [59].
In histone H3: K4 (lysine at position 4); M, methyl (me); me3, timethyl -> H3K4me3
Thank you again for your input.
Minou Bina

Reviewer 2 Report
1. Dear author, am I correct in assuming that a valid exposure is counted from the value of 3 peaks? I don't quite understand how this threshold was derived. Could you please explain or point me to the correct line? (line 137) 12. You have very exciting work on this topic. It would be interesting to know if there were any validation practical lab work to clarify the result? (line 61)Author Response
Thank you for your time spent for reviewing my manuscript. Below, with -* I have marked your
questions.
-* Dear author, am I correct in assuming that a valid exposure is counted from the value of 3 peaks? I
don't quite understand how this threshold was derived.
Response: I selected the window size by trial and error. For windows larger than 850 bases, I noticed
spurious peaks producing false positives. In windows shorter than 850 bases, peaks appeared spiky.
A?er creating the plots, I closely inspected peak positions with respect to well characterized gDMRs/ICRs
in mouse. I found that peaks encompassing 3 or more ZFBS-morph overlaps accurately located known
gDMRs/ICRs in mouse. In rare instances, I observed a peak near a nonimprinted gene. For example: in
Herc3 locus, a peak covering 2 ZFBS-morph overlaps accurately predicted the intragenic ICR regulating
imprinted Nap1l5 repression. However, I also noticed a peak near Herc3 (not an imprinted gene).
Therefore, I concluded that a peak covering 2 ZFBS-morph overlaps could be true or false positive. In
Bos taurus, I did not find a false positive in the HERC3 locus.
-* Could you please explain or point me to the correct line? (line 137) 12. You have very exciting work on
this topic. It would be interesting to know if there were any validation practical lab work to clarify the
result? (line 61)
Response: Sorry, I don’t understand your question about “point me to the correct line? (line 137) 12.”
Thank you for finding my work exciting. I really appreciate it.
I do not have any experimental validations. However, since my genome wide studies located known ICRs
in three different mammalian species (mouse [46], human [47], and Bos taurus [48]), I concluded that
my strategy made correct predictions. Please note that discovery of new imprinted genes is extremely
expensive and time consuming. My results could guide researchers where to look in genomic DNA to
find potential imprinted genes for experimental validation. For example: researchers interested in bull
fertility could design experiments to test whether CNNM1 and CNR1 are imprinted genes. Researchers
interested in beef production could design experiments to test whether SIX1 and BCL6 are imprinted
genes. I don’t have a wet lab to perform experiments. However, I could imagine that with availability of
SNPs, one could design primers to distinguish the maternal and paternal alleles. In that case, available
technologies could be used to determine whether a predicted ICR was methylated in one of the two
alleles.
Thank you again for your input.
Minou Bina

Round 2
Reviewer 1 Report
No comments.